# Identification of Therapeutic Targets for Medulloblastoma by Tissue-Specific Genome-Scale Metabolic Model

**DOI:** 10.3390/molecules28020779

**Published:** 2023-01-12

**Authors:** Ilkay Irem Ozbek, Kutlu O. Ulgen

**Affiliations:** Chemical Engineering Department, Bogazici University, Bebek, Istanbul 34342, Turkey

**Keywords:** medulloblastoma, systems biology, metabolic brain model, sphingolipid metabolism, drug target

## Abstract

Medulloblastoma (MB), occurring in the cerebellum, is the most common childhood brain tumor. Because conventional methods decline life quality and endanger children with detrimental side effects, computer models are needed to imitate the characteristics of cancer cells and uncover effective therapeutic targets with minimum toxic effects on healthy cells. In this study, metabolic changes specific to MB were captured by the genome-scale metabolic brain model integrated with transcriptome data. To determine the roles of sphingolipid metabolism in proliferation and metastasis in the cancer cell, 79 reactions were incorporated into the MB model. The pathways employed by MB without a carbon source and the link between metastasis and the Warburg effect were examined in detail. To reveal therapeutic targets for MB, biomass-coupled reactions, the essential genes/gene products, and the antimetabolites, which might deplete the use of metabolites in cells by triggering competitive inhibition, were determined. As a result, interfering with the enzymes associated with fatty acid synthesis (FAs) and the mevalonate pathway in cholesterol synthesis, suppressing cardiolipin production, and tumor-supporting sphingolipid metabolites might be effective therapeutic approaches for MB. Moreover, decreasing the activity of succinate synthesis and GABA-catalyzing enzymes concurrently might be a promising strategy for metastatic MB.

## 1. Introduction

Medulloblastoma (MB) is the most common brain tumor in children with an incidence of five per million [1,2]. MB is an embryonal tumor forming in the cerebellum and is estimated to be derived from the neuronal progenitor or stem cells [2]. Based on the unique molecular and clinical properties of MB types, it was classified into four subgroups WNT, Sonic Hedgehog (SHH), Group 3 (GR3), and Group 4 (GR4) [3]. The standard conventional treatment procedure applied to the patients is radiation and chemotherapies following the removal of the tumor [4]. Children younger than three do not receive radiation therapy as it may cause permanent damage to the developing brain, consequently, this may put their survival at stake [4]. Furthermore, many patients experience permanent neurological sequelae such as a decrease in intelligence, memory impairment, and trouble concentrating because of these therapies [5,6]. Hence, there is a vital need for computational metabolic models that can imitate medulloblastoma characteristics in order to estimate therapeutic targets with minimum toxic effects on healthy cells and thus increase the patient’s life quality. Genome-scale metabolic models connecting the gene, reaction, and metabolite have allowed researchers to examine cancer metabolism over the past few decades [7,8,9]. 

Metabolic changes stemming from medulloblastoma include the over-expression of enzymes related to lipid production and the Warburg effect in which abnormal glucose consumption and overproduction of lactate are observed regardless of oxygen presence [10,11]. Reduction in the activity of the oxidative phosphorylation (OXPHOS) pathway is another typical trait of the Warburg effect. Still, some studies reported that MB cells also utilize OXPHOS to support their migration and survive in different conditions. 

Furthermore, MB cells were found to upregulate the amino acid transporter SLC1A5, which takes up glutamine in cells [12]. High activity was also observed in both non-oxidative and oxidative branches of the Pentose Phosphate Pathway in MB cells to support nucleotide production, which is necessary for their rapid proliferation [13,14]. 

In the present work, the genome-scale brain metabolic model iMS570 reflects healthy brain metabolism and was constructed previously by our group [15] and then changed to the iMS570g model to capture glioblastoma-specific properties [9], was modified and expanded by taurine and sphingolipid metabolism reactions. The revised model, integrated with MB-specific transcriptome data, was named the MB model and the metabolic alterations in medulloblastoma were simulated including those of metastatic MB cases. Flux Balance Analysis (FBA) combined with Minimization of Metabolic Adjustment (MOMA) were used to analyze the MB cases under several conditions. The reason for the dual objective function in this work is that it has predicted more accurate results in human brain metabolism in previous studies due to its minimum enzyme use to obtain the objective specified for FBA [9,16]. The regulation, Flux Coupling (FCA) and essentiality analyses were conducted to uncover potential therapeutic targets in life-threatening MB.

## 2. Results

The genome-scale metabolic brain models are reconstructed for healthy and MB cases (nine MB models) and integrated with transcriptomic data (three GSE data incorporated with Gene Inactivity Moderated by Metabolism and Expression (GIMME)). Two MB models were reconstructed by using the average of all gene expression levels in GSE62600 and GSE37418 in order to compare the results with each other and with the healthy results. Four MB-specific models that reflect subtypes of MB were created and named WNT, SHH, GR3, and GR4 using the GSE37418, including gene expression levels of these subtypes. Lastly, the non-metastatic (M0) model, metastatic (M2), and (M4) MB models were developed employing GSE10327 to find the relation between the Warburg effect and metastasis. The models and simulation results were validated by comparing the FBA results with the data obtained experimentally in the literature. The main reactions carried out in the cell are presented in Figure 1. 

The substantial metabolic differences between the MB and healthy cases and the experimental findings in the literature are summarized in Table 1. These results in Table 1 were obtained from MB, SHH, WNT, GR3 and GR4 models where the GSE37418 dataset is utilized. The detailed results acquired from these models and other MB models created using the GSE62600 and GSE10327 datasets are available in Appendix A. The most striking metabolic change that stands out in MB is the increase in lactate production due to the Warburg effect, where MB cells prefer glycolysis over oxidative phosphorylation to generate ATP regardless of oxygen quantity [10]. Overproduction of lactate and low aerobic respiration activities were predicted by all MB-specific models, consistent with the study, in which cerebellar granule cell with Sonic Hedgehog (SHH) produces twice as much lactate as its glucose uptake (R593 + R594) [17]. Since cancer cells require ribose-5 phosphate for nucleotide production [18], flux rates in the non-oxidative phosphate pathway were observed to increase. 

Despite low TCA flux rates in the MB models, acetyl-CoA formation (R28 + R72) in the TCA cycle, which is necessary for lipid synthesis, is higher compared to the healthy brain model. Acetyl-CoA is produced in the pyruvate dehydrogenase reaction (R13 + R57) and threonine metabolism (R197 + R200). The glutaminase reaction (R96) showed higher activity in all MB models than the healthy model, which is consistent with the literature where the low expression levels of glutamate uptake proteins, EAAT1 and EAAT2-4 were observed [12]. Glutamine-derived glutamate is converted to alpha-ketoglutarate in a glutamate dehydrogenase reaction (R89 + R90 + R92 + R93) to feed the TCA cycle. Consistent with the literature, flux rates of the glutamate dehydrogenase reaction (R89 + R90 + R92 + R93) were observed to be active in the MB model (see Appendix A). 

The ratio of glutamate over glutamine accumulation in the cerebellum predicted by the MB model was found to be very close to the concentration ratio determined experimentally [21] (Table 1). 

### 2.1. Energy Production and Growth Rate without Glucose and Glutamine Sources

When glucose is available, the MB cells mainly use aerobic glycolysis because of the Warburg effect. Even though OXPHOS was also utilized by MB cells, the importance of glycolysis for both energy production and cell proliferation is undeniable. Additionally, due to the heterogeneity of the tumor environment, tumor cells may require different metabolic needs. Using FBA, the glucose uptake rate was restricted to zero to be able to observe the impacts of glucose deficiency on energy production in the MB. MB, which utilizes glycolysis regardless of oxygen availability, was observed to be almost entirely reliant on OXPHOS when glucose is not available. In this case, whole energy generation was reduced by 82% (Figure 2a) and the cell received its energy demands mainly from OXPHOS (Figure 2b). 

Then, to simulate the effects of a glutamine deficiency, two consecutive adjustments to the MB model, glutamine uptake and both glutamine and glucose uptake rates, were done where these uptake rates were reduced to zero.

In the absence of glutamine, energy generation in the TCA cycle decreased to zero. In both glutamine and glucose deficiencies, energy generated in the TCA cycle declined from 6% to 4% (Figure 2b). In both cases, MB obtained most of its ATP requirements from OXPHOS. However, glutamine deficiency did not affect the cell as much as the absence of glucose since the total ATP generation rate was decreased by only 20% in the case of glutamine deficiency. Still, in three cases, MB utilized OXPHOS, which was the only possibility for energy generation. In the healthy model, the same analyses were repeated. However, FBA for glucose deficiency in the healthy model was found to be zero because of the reduced solution space. Therefore, it is reasonable to conclude that the absence of glucose uptake has a huge impact on healthy cells. On the other hand, as found in the MB model, the absence of glutamine uptake seems to have no effect on the healthy model. 

Then, biomass rate (R25) was also examined in cancer cells to investigate the impact of carbon source deficiency on the growth of MB cells (Figure 2c). It was observed that without glutamine, MB cells can survive since the biomass rate remained constant. However, glucose deficiency decreased the biomass rate to zero, which indicates that glucose is crucial for MB cells to proliferate. Cancer cells activate the PPP to promote their growth by producing ribose-5 phosphate, which is significant for nucleotide and NADPH synthesis. In agreement with this, ribose-5 phosphate production rates (R21 + R65) decreased from 0.021 mmol/gDW/h to zero (see Appendix A). In the MB model, in the case of glucose and glutamine deprivation, threonine uptake (E621) drastically increased (from 0.00003 mmol/gDW/h to 0.05 mmol/gDW/h) to replenish the TCA cycle and OXPHOS by transforming to cysteine and acetyl-CoA. The reduction in total energy generation detected in both cases signals that the glucose deficiency mitigates the probability of survival for MB but also affects the healthy cells. 

### 2.2. The Relationship between Metastasis and the Warburg Effect

The MB-M0, MB-M2, and MB-M4 models, which mimic non-metastatic, Grade 2 and Grade 4 metastatic MB tumors, were created by incorporating the MB model with the GSE10327 dataset, which contains gene expressions of 62 human medulloblastoma patients [22]. Due to the higher expression levels of the *Hexokinase 2* (*HK2)*, *Phosphofructokinase* (*PFK)*, and *Pyruvate Kinase* (*PK)* genes in metastatic samples, the ratio of ATP produced in glycolysis over ATP produced in OXPHOS (ATPG/ATPOP) was detected to be almost seven times higher in MB-M2 than the ATPG/ATPOP found in MB-M0 (Figure 3). 

Even though the remarkable change in ATPG/ATPOP estimated between MB-M0 and MB-M2 was not observed between MB-M2 and MB-M4, ATPG/ATPOP was still higher in MB-M4 than the one in MB-M2. It was noted that while the ratio of ATP produced in glycolysis over the energy produced in OXPHOS rises, migration augments in the NCI-60 cell lines [23]. However, while metastasis increased, the ratio of lactate excretion to oxygen uptake rate (LacR/OCR) enhanced only slightly. Still, the difference detected in ATPG/ATPOP suggests that the Warburg effect rises with the metastatic grade in MB. Metastasis in MB might be suppressed by interfering with the genes related to the Warburg effect and decreasing ATPG/ATPOP. 

### 2.3. Flux Sampling and Regulation of Reactions

By using the Flux sampling method, 10 000 flux profiles that satisfy the constraints were attained for each reaction. There are 594 common reactions in both healthy and MB models (Appendix A). Based on the *p*-values obtained and adjusted using a two-sample t-test with unequal variances and the Benjamini–Hochberg (BH) procedure, 205 reactions out of 594 were estimated to be significantly different from one another. In the MB neuron cells, 73% of reactions (11 out of 15 reactions) in the glycolysis pathway showed statistically important deviation from the healthy neuron cells because of the increased glucose uptake and excessive lactate production. As was observed in the FBA result, lactate production was determined to be statistically higher relative to the outcome for the healthy model (see Appendix A). Energy generation in OXPHOS was also found to be notably decreased in the MB model in comparison to the healthy model (see Appendix A). In agreement with FBA and the experimental results, 80% of the OXPHOS pathway reactions (four out of five reactions) were determined to be significantly different for the neuron cells of MB relative to the neurons of the healthy model because of the Warburg effect.

For regulation, Z scores were estimated and named ZF and ZG, which refer to a statistical change in reaction fluxes and gene expression levels, respectively, in both cases. The reactions with metabolic fluxes (ZF) and gene expression data values (ZG) that were higher than 1.96 or lower than −1.96 were categorized as transcriptionally regulated reactions and reactions that are not regulated at the transcriptional level (NTR). The reactions where both flux and gene expression alter are called transcriptionally regulated (TR) [24]. As a result of this analysis, 80 reactions were identified to be controlled at the transcriptional level and 313 reactions were found not to be regulated at the transcriptional level. These results are available in Appendix A. Figure 4 shows the number of reactions controlled at the transcriptional level and the reactions that are not controlled at the transcriptional level in both cells for each pathway.

In glycolysis, most of the reactions are not regulated at the transcriptional level (Figure 4). This stems from the fact that a high amount of glucose taken by glucose carriers activates the enzymes catalyzing the reactions in the glycolysis pathway.

It was detected that lactate dehydrogenase in the glycolysis pathway has an increased ZF score, although its ZG score is remarkably low. A high level of pyruvate might be causing an increase in lactate formation.

Fatty acid synthesis had the highest TR ratio with more than one-third of reactions being transcriptionally regulated (Figure 4). 

The *ACOT7* gene controls the stearate production from stearoyl-CoA and *FADS2*, and *SCD*, the genes that regulate oleoyl-CoA production from stearoyl-CoA. Due to significantly low ZF and ZG scores for the *ACOT7*, *FADS2*, and *SCD* genes, it is pertinent to point out that the oleoyl-CoA production is downregulated in MB and there is a decrease in oleic acid quantity in MB relative to the healthy brain. Oleoyl-CoA, which is the metabolically active form of oleic acid, has anti-cancer features [25,26]. Based on the literature results acquired with the Raman technique, oleic acid production was downregulated in MB [27]. Therefore, interfering with these genes, which play essential roles in oleic acid production, might be a promising therapeutic strategy.

### 2.4. Drug Targets for Medulloblastoma

#### 2.4.1. Targeting Biomass-Coupled Reactions to Inhibit Tumor Growth

The reactions related to biomass reactions that do not affect the vital functions of healthy cells are counted as potential drug targets to suppress tumor growth. The reactions controlling the uptake of essential amino acids including phenylalanine, lysine, and histidine [28] were fully connected to growth reaction, indicating that targeting these reactions affects tumor growth significantly. For example, an extensive amount of lysine uptake was reported for breast cancer cells [29]. Lysine-free diets were also suggested to assist in cancer therapy [30]. The LAT1 protein takes up isoleucine, histidine, leucine, and phenylalanine and regulates the reactions of R598, E25, R597 and R603 in the MB model. The LAT1 protein was overexpressed in MB cases [12,31]. Additionally, the inhibition of the lysine uptake reaction did not affect the vital functions of healthy cells significantly. These experimental outcomes and the findings of this study suggest that the inhibition of essential amino acid internalization reactions on MB should be examined in further works. Another flux coupled to biomass is the one in which glutamate and asparagine are synthesized from glutamine and aspartate (Figure 5). The *asparagine synthetase*(*ASNS)* gene regulating this reaction was upregulated in some cancer types such as pancreatic, ovarian, and prostate [32]. Reduced asparagine quantity in brain metabolism improved the effect of chemotherapy remarkably in the DAOY, MB cell line by enhancing the sensitivity of MB cells [32]. Thus, *ASNS*, which was found by the MB model in both analyses, could be a potential therapeutic target for MB.

The reactions associated with the production of cardiolipin, which is used in the structure of the cell membrane [33], are directly coupled to the biomass reaction (Figure 5). Cytidine diphosphate diacylglycerol (CDP diacylglycerol) and glycerol-3-phosphate [15] are converted to cardiolipin. Glycerol-3-phosphate production reactions are found in flux coupling analysis as this intermediate is utilized in both CDP diacylglycerol and cardiolipin formations. CDP diacylglycerol is the key metabolite in cardiolipin and phosphatidyl-inositol syntheses [34]. Consistent with these data, the reactions in CDP diacylglycerol metabolism were determined as reactions partially coupled to biomass (Figure 5) and may serve as putative drug targets against MB.

Consistent with other analyses performed in this work, the reaction where cholesterol produced in an astrocyte is transferred to the neuron was also found to be directly coupled to the growth reaction (R25).

#### 2.4.2. Genes and Gene Products as Targets for Suppressing Tumor Growth: Essentiality Analyses

Single/double gene and reaction deletion analyses were performed to be able to find potential therapeutic targets for MB. In essentiality analysis, eight genes out of forty-two were removed as they significantly affected the main functions of the healthy cells (see Appendix A). When these genes were eliminated, the single-gene and reaction deletion analyses indicated that 68% of essential genes and reactions were associated with the main lipid metabolism. 

The essential reactions were found in pathways such as phosphatidylethanolamine, cholesterol, sphingomyelin, glycosphingolipid, phosphatidyl-inositol, and fatty acid metabolism. In agreement with the experimental findings in which lipid was found abundantly in MB [20], lipid production was considered crucial for MB cell proliferation.

Many of the controlling genes/gene products related to the reactions detected in flux coupling analysis were also found in essentiality analyses. For instance, *ASNS*, which is associated with several malignancies [32], was identified in gene essentiality analysis as an essential gene. In addition, the cardiolipin synthesis genes, *PGS* and *CRLS1*, and the glycerol-3-phosphate syntheses genes *GPD1* and *GPD2* were found as essential genes. Glycerol-3-phosphate is utilized in cardiolipin metabolism to produce cardiolipin, making up 20% of the mitochondrial membrane [33]. *GPD2* was observed to be excessively active in numerous cancers [35]. Furthermore, *GPD2* inhibition led to anti-cancer effects in a prostate cancer cell line [36]. Consistently, *GPD1* and *GPD2* genes and the reactions controlled by them were found in both flux coupling and single-gene deletion analyses by the MB model. Furthermore, inhibition of these genes did not cause severe changes in the vital functions of the healthy model.

The *ST3GAL5* gene regulating GM3 synthesis was identified as essential with the MB model and GM3 was found abundantly in MB by Chang et al. [37]. Even though GM3 showed apoptotic features in many cancer types, suppressing GM3 synthase led to anti-cancer effects in MB [38,39]. Particularly, in the Group 4 type of MB, the *ST3GAL5* gene was highly active in comparison to the control group and other subtypes [40]. Hence, an analysis specific to the MB-GR4 model was carried out. Silencing GM3 synthase led to a 95% reduction in the growth of the GR4 type of tumor.

Another essential gene found by the MB model was the *B4GALNT1* gene that controls GM2, GD2, and GT2 syntheses using GM3, GD3, and GT3 gangliosides, respectively [41]. The most and the second most abundant ganglioside in the DAOY and TE-671 MB cell lines, respectively, is the GM2 ganglioside, which induces tumor proliferation [37,41,42]. The GD2 level was also found to be notably higher in MB relative to the healthy brain [43]. Only MB was observed to have a high amount of GD2 among other childhood cancers [44].

In double gene deletion analysis, 26 synthetic lethal gene pairs were determined once the genes found in single deletion analysis were removed (Figure 6). However, 10 synthetic lethal gene pairs were removed because their inhibition affected the main functions of the healthy model significantly (see Appendix A).

The combination of the *ATP citrate lyase* (*ACL)* and *Fatty Acid Synthase* (*FASN)* genes was captured by the MB model in double gene deletion analysis. In agreement with our simulations, simultaneous inhibition of the *ACL* and *FAS* genes hindered cancer cell division [45]. Moreover, silencing *ACL*, which is responsible for the production of acetyl CoA in the TCA cycle, reduced the growth of lung cancer cells [45]. Suppressing *FASN* also caused a decrease in the hallmarks of tumors such as lipid production in various cancers such as pancreatic, ovarian, and lung tumors [46]. Although such an approach is toxic to healthy cells considering the importance of these two genes for the cell, the MB model successfully identified two genes whose deletion could be destructive also for MB.

Combinations of the *ABAT* gene with *succinate-CoA ligase ADP-forming subunit beta* (*SUCLA2*) and *succinate-CoA ligase GDP/ADP-forming subunit alpha* (*SUCLG1*) genes that regulate succinate synthesis (R34 + R77) from succinyl-CoA in the TCA cycle were also determined by the MB model (Figure 7). The activity of *GABA transaminase* (*ABAT*), which controls SuccinateSAL and glutamate productions (R99 + R101) from alpha-ketoglutarate and GABA, is relatively lower in primary MB than in healthy cerebellum since overexpression of *ABAT* in MB decelerates proliferation and increases the activity of OXPHOS [12,47]. In fact, in more malignant types of MB, in GR3 and GR4 subtypes, *ABAT* was found to be less active than SHH and WNT. Furthermore, activation of *ABAT* was observed to trigger apoptosis and diminish cancer cell division in rats with MB [47]. However, recently, *ABAT* expression was found to be enhanced in metastatic MB in comparison to primary MB in order to meet its energy needs by utilizing GABA [47]. That is, metastatic tumor cells were detected to rely on OXPHOS to obtain enough energy and to withstand hard conditions. In addition, the *ABAT* gene was found to be crucial for metastatic cells because *ABAT* recharges the metabolites necessary for the TCA cycle and OXPHOS by contributing to succinate and NADH productions.

Succinate upregulation is also associated with cancer [48]. Abnormalities associated with succinate dehydrogenase (*SDH*), which converts succinate to fumarate, were found in some malignancies as high succinate levels cause hypoxia-inducible factor 1-alpha (HIF-alpha) stabilization and HIF-alpha controls the genes triggering cell proliferation [48]. However, it is also important to note that when these two synthetic lethal gene pairs were inhibited in the healthy model, lower OXHOS activity was observed due to low GABA and TCA activity and ATP production capacity decreased by 70%. Therefore, even though interfering with the genes that control succinate synthesis has potential in terms of cancer treatment, further research is needed to determine the correct dose of drugs to minimize the toxic effect on healthy cells or to develop an approach that targets only cancer cells.

In sphingolipid metabolism, the *SGPL1* gene, which regulates reversible phosphoryl-ethanolamine synthesis from sphingosine 1-phosphate (SP1), was identified with the *KDSR* and *DEGS2* genes (Figure 1). While ceramide triggers cell apoptosis, S1P promotes cell survival, hence cancer cells form S1P from ceramide to refrain from cell arrest [49,50]. Thus, the MB model found the *SGPL1* gene whose activation might result in the formation of sphingosine 1-phosphate even if the *DEGS2* or *KDSR* genes were silenced. Because both sphingomyelin and GM3 are produced from ceramide, targeting the genes that regulate their syntheses might lead to the accumulation of ceramide, which induces apoptosis [51]. Compatible with these data, the sphingomyelin synthesis genes, *SAMD8*, *SGMS1*, and *SGMS2* genes, together with the *ST3GAL5* gene-regulating GM3 production were determined to be essential by the MB model.

#### 2.4.3. Targeting Common Essential Genes in Biomass, Lactate, and Energy Productions

Single gene essentiality was repeated for lactate and energy productions as the objective functions to capture the genes that are crucial for three parameters (biomass, lactate, energy) since interfering with these genes was considered to improve the therapeutic strategy. In total, 32 common essential genes were found in the three analyses. Then, these 32 essential genes were silenced one at a time from the normal brain model to identify the ones whose knockout does not affect the necessary functions of the healthy brain cells. The genes that decrease MB growth by more than 40% were determined. Taking these criteria into consideration, the essential gene list was narrowed down to eight. By using the DrugBank and GeneCard databases, 54 drugs/compounds were found for eight essential genes (*HMGCR*, *MVK*, *PMVK*, *MVD*, *FDFT1*, *CRLS1*, *FASN*, and *SQLE*).

The first six genes regulate the mevalonate pathway, which is a therapeutically promising pathway in terms of cancer. The MB model successfully assisted us in identifying 24 out of 54 medications that are linked to cancer through their targets.

Statins, which are used in cholesterol and cardiovascular issues, suppress *HMGCR*, which is responsible for mevalonate synthesis [52]. One of the statins, Lovastatin, was investigated as a possible cancer drug in some studies linked to glioblastoma, astrocytoma, and breast cancer [53,54]. Lovastatin also mitigated cell growth and triggered apoptosis in MB cells [55]. Another statin, Simvastatin, was found to induce apoptosis in MB cell lines [56]. Another cholesterol-related drug, TAK-475, works by deactivating squalene synthase [57]. Decreasing squalene synthase activity enhanced the susceptibility of cancer cells to chemoimmunotherapy [58]. By targeting fatty acid metabolism, tetrahydrolipstatin (Orlistat—a drug to treat obesity) deaccelerates the cancer progress [59]. Orlistat decreased growth by 64% and depleted cell viability in glioblastoma cells [59].

Cerulenin is an antifungal agent that hampers sterols and fatty acid syntheses and reduces the activity of HMG-CoA synthase in cholesterol synthesis [60]. Cerulenin was shown to induce apoptosis in MB [61].

Ellagic acid (EA) was studied in numerous tumors such as colon cancer, lung cancer, ovarian cancer, melanoma, and GBM [62]. Studies realized in GBM uncovered that the tumor-suppressive feature of EA was linked to the inactivation of the Akt and NOTCH1 signaling pathways whose controlling genes are upregulated in MB [62]. NOTCH1 is the primary reason for Group 3 MB metastasis [63].

#### 2.4.4. Antimetabolites Competing with Natural Metabolites

Antimetabolites are chemically similar substances to natural metabolites and they are often used for cancer and viral diseases to inhibit the use of natural metabolites [64]. To identify potential antimetabolites for MB, drug effects were simulated for each metabolite in both models. Once the reactions where these metabolites are reactant were identified, these related reactions were inhibited one by one to create competitive inhibition. The metabolites consumed in the reactions whose enzyme inhibition does not affect the main functions of healthy cells significantly and decrease MB cell growth by more than 40% were determined. As a result, 114 natural metabolites in reactions, whose enzymes could be used as therapeutic targets, were captured by the healthy and MB models. These metabolites are available in Appendix A. In total, 32% of the substrates detected by both models are related to FAs. Because they are used in a variety of pathways that are crucial for cancer progression, FAs were investigated as a target for a number of diseases including several cancers [65]. Tumor cells employ acetate or glucose to form lipids. In the SHH subgroup of MB, *FASN* was determined to be overexpressed [10].

Metabolites of cholesterol metabolism were found to make up 10% of the total detected substrates. These metabolites are mostly intermediates of the mevalonate pathway, which is stimulated by malicious cells to support cancer-associated features such as uncontrollable growth [66]. In addition to statins which target the mevalonate pathway [52], Meglutol, which inhibits *HMGCR*, was found to be considerably similar to mevalonate with a 0.91 Tanimoto score; however, it crosses the blood–brain barrier with poor transport capacity [67].

Fifteen percent of natural metabolites consumed in reactions whose catalyzing enzymes are potential targets are linked to sphingolipid metabolism. While 8% of these metabolites were determined to be associated with the sphingomyelin pathway, 7% of them are related to the glycosphingolipid pathway. Indeed, enzymes and intermediates associated with sphingolipid metabolism were determined to play vital roles in cell signaling mechanisms. One of the antimetabolite candidates detected similar to sphinganine metabolite is phytosphingosine with a 0.95 Tanimoto score. Sphinganine is used in the production of sphinganine-1-phosphate and dihydroceramide [51]. Phytosphingosine, which is an analog of sphingosine, is found in white blood cells and microvilli in the small bowel of mammals [68,69,70]. Sphinganine and sphingosine were determined to stimulate apoptosis in colorectal cancer [71]. Furthermore, sphinganine triggered apoptosis in leukemia cells [72]. Phytosphingosine stimulates apoptosis, releasing cytochrome c, which leads to the stimulation of caspase 3 in lung cancer and lymphoma cells [69,70]. Further studies demonstrated the therapeutic impacts of phytosphingosine on cancer cells [73,74]. Phytosphingosine suppresses ERK1/2, a survival pathway, and supports p38 MAPK, which triggers cytochrome c secretion [70]. In a subsequent study on lymphoma, phytosphingosine and γ-radiation simultaneously initiated apoptosis of the cells resilient to radiation [70].

Despite its anti-cancer features, its delivery might be challenging owing to its low solubility and permeability through the blood–brain barrier (BBB) [67,75]. The phytosphingosine–liposome composition was developed to increase the penetration of phytosphingosine through the skin [75]. Recently, cationic phytosphingosine nanoemulsions were employed to deliver plasmid DNA to breast cancer cells [76].

Considering the lack of studies on phytosphingosine delivery, further research using nano-engineering techniques including micelles, nano-capsules, liposomes, and emulsions seems promising.

## 3. Discussion

Among pediatric brain tumors, medulloblastoma (MB) is the most prevalent kind (constituting nearly 20 percent of all pediatric brain tumors) with an annual rate of ~5 cases per one million population [1,2]. The treatment for MB is surgery followed by radiation and chemotherapy to remove any remaining tumor but these have dramatic side effects such as a decline in intelligence, low survival rates, etc. In the present work, metabolic changes specific to medulloblastoma were captured by employing system biology tools, and potential therapeutic strategies for MB were proposed based on the outcomes of a tissue-specific genome-scale metabolic modeling study. The MB model consists of 753 metabolic reactions modulated by 601 genes and 44 pathways along with glycosphingolipid metabolism and taurine synthesis, and the biomass reactions established based on the white matter were customized specifically to the cerebellum. As a result of performing both flux balance analysis with MOMA and flux sampling analysis, the activity in glycolysis was notably higher in MB-specific models in comparison to the healthy model (see Appendix A for flux sampling histograms). On the contrary, the flux values in the TCA and OXPHOS pathways were much less than those in the healthy model in accordance with the literature. Furthermore, the reactions that promote lipid and nucleotide productions in the TCA cycle and PPP, respectively, were much more active than those in healthy cases. Hence, the MB model predicted metabolic phenotypes of MB effectively as validated by experimental works.

In an attempt to find the drug targets for MB, 158 reactions were estimated to be coupled with the growth reaction. Interfering with these reactions may be a potential therapeutic strategy for MB. Because various cancer agents are interfering with nucleotide metabolism and considering how the MB model captured nucleotide exchange reactions coupled to growth reaction, the MB model made the right predictions about cancer metabolism. In addition, reactions associated with the production of CDP diacylglycerol and glycerol-3-phosphate and main cardiolipin production reactions stand out in the flux coupling analysis. Moreover, cardiolipin and glycerol-3-phosphate synthesis reactions and the genes regulating them (*CRLS1*, *PGS1*, *GPD1*, and *GPD2*) were determined as essential in the essentiality analyses. Notably, the *CRLS1* gene was also identified as one of eight essential genes required for ATP and lactate production besides growth. In the essentiality analyses, 45 genes and 312 reactions were identified as essential. A total of 42 of them were detected in both analyses and almost 60% of the essential genes were associated with lipid production. In sphingolipid metabolism, the *B4GALNT1* and *ST3GAL5* genes were determined as essential by the MB model. The fact that inhibition of GM3 synthase led to anti-cancer effects in MB, and GM3 was found abundantly in MB, renders the *ST3GAL5* gene a desirable target for MB treatment. The essentiality analyses for biomass, energy and lactate productions resulted in eight common genes (*HMGCR*, *MVK*, *PMVK*, *MVD*, *FDFT1*, *SQLE*, *CRLS1*, and *FASN*), regarding their toxic effects on the healthy model and their potential as therapeutic targets for MB. The antimetabolites that result in competitive inhibition to natural substrates in reactions were discovered, and it was found that 32%, 15%, and 10% of natural metabolites in reactions whose enzymes can be used as therapeutic targets were related to fatty acid synthesis, sphingolipid metabolism, and cholesterol synthesis, respectively. Mevalonate–meglutol and sphinganine–phytosphingosine were identified to be very similar to each other with 0.91 and 0.95 Tanimoto scores, respectively. Phytosphingosine has anti-cancer properties and has the potential to act as an antimetabolite, but further experimental study is needed to fully understand how it enters the BBB.

This study provides a systematic approach for the identification of drug targets using system biology tools, which link constituents of biological systems and allow scientists to study the interactions of biological components by taking advantage of computational and experimental methods. Researchers can use these putative targets for MB, the embryonal tumor that develops in the cerebellum early in life, as a roadmap to thoroughly investigate them experimentally.

## 4. Materials and Methods

### 4.1. Reconstruction

iMS570, which includes 630 metabolic reactions carried out between astrocytes and neurons controlled by 570 genes [15], covers essential pathways such as glycolysis, pentose phosphate pathway, citric acid cycle, oxidative phosphorylation, and lipid production pathways. The objective function used in iMS570 was the maximization of the sum of glutamate/glutamine/GABA cycle flux rates. iMS570g, which contains 659 reactions regulated by 572 genes, also has biomass reactions based on white matter where glioblastoma occurs and four reactions related to glutaminolysis metabolism. With the addition of new reactions, our MB model covers 753 metabolic reactions regulated by 601 genes (Appendix A). The set of growth reactions found in iMS570g was amended based on the composition of the brain cerebellum where MB originated [2,9,77,78,79,80,81,82,83,84]. Since high amount of taurine is one of the prominent features of medulloblastoma, 15 taurine synthesis reactions [20,85] were added to the model. Additionally, sphingomyelin pathway was expanded and glycosphingolipid metabolism was added to the MB model in order to investigate the effect of these pathways on brain cancer metabolism. Therefore, 79 reactions covering phosphoryl-ethanolamine, sphingosine-1-phosphate (S1P), and ceramide synthesis reactions in sphingolipid and glycosphingolipid metabolism pathways were incorporated into the MB model. Furthermore, galactose uptake reactions were added because galactose is utilized in the production of galactosylceramide [86]. The distribution of neurons and astrocytes in the cerebellum was determined as 97% and 3%, respectively [87]. Thus, uptake rates defined as constraints were shared between two cells based on this proportion. In MB models, metabolites whose names include N and A letters are internalized and/or used by neurons and astrocytes, respectively. Metabolites accumulated in the cerebellum are shown with the letter ‘C’ next to their names. The related codes and transcriptomic data used for model reconstruction, are available in Appendix A.

#### 4.1.1. Transcriptomic Data Integration

Medulloblastoma-specific genome-scale models were generated by integrating three GSE datasets from two platforms. The data in GSE62600 [40] and GSE37418 [88] from GPL96 and GPL570 platforms, respectively, were categorized according to MB subgroups. While GSE62600 was used to compare MB with the healthy case, GSE37418 was employed to make comparisons between MB subgroups. The third dataset, GSE10327 from the GPL570 platform containing metastatic, and non-metastatic MB samples, was utilized to reveal the relationship between the metastasis and the Warburg effect. These datasets were incorporated into the MB model by GIMME [89], which employs binarized gene expression data and a genome-scale metabolic model. Each gene in the datasets was classified as Absent (A) and Present (P) according to its gene expression levels. The genes whose expression level was below a specified value were designated as A, while the ones with higher expression levels than this value were assigned to P. For this procedure, the threshold values were determined for each MB-specific model to identify the overexpressed and lower-expressed genes. 

The expression levels of *HK2*, *Pyruvate kinase M2 subtype* (*PKM2*), *FASN*, *Acetyl-CoA Carboxylase 1* (*ACACA*), *Glutaminase* (*GLS*) were the upper limit for the threshold value since these genes are upregulated in MB [10,12,17,45,85,90,91]. Based on this criterion, 72.3% of the average of data expression levels of each gene in the GSE37418 dataset was selected. That threshold was barely less than the *FASN* expression value and high enough to create a model which reflects the MB characteristics.

To generate specific models for MB subtypes, four different thresholds were created by using averages of gene expression of patients in different subgroups. Taking into consideration the criteria determined, thresholds were chosen as 72.5%, 74.7 %, 90.24%, and 76.34% of the average of WNT, SHH, GR3, and GR4 gene expressions, respectively. GIMME deleted 7, 7, 9, 13, and 8 reactions from MB, WNT, SHH, GR3, and GR4 models, respectively.

The differences between gene expression values in GSE62600 were higher compared to GSE37418. Thus, small variations in the threshold did not influence the results remarkably. The thresholds that are more than 50% of the transcriptome data average of MB were eliminated because these values were exceeding the *GLS* expression value. As in GSE37418, the expression levels of upregulated genes in MB were determined as an upper bound. Consequently, 30% of the transcriptome data values average was selected, and 38 reactions were erased by GIMME from the MB model for GSE62600.

To investigate the correlation between metastasis and the Warburg effect, the expression values of M0, M2, and M4 medulloblastoma metastasis stages that are present in GSE10327 were utilized to construct MB models that reflect metastasis-specific properties. As thresholds, 45%, 50%, and 55% of the gene expression averages were selected for the non-metastatic (M0), M2, and M4 stages, respectively. GIMME eliminated 31, 31, and 26 reactions from the non-metastatic (M0) model, metastatic (M2), and (M4) MB models, respectively.

#### 4.1.2. Constraints

The context-specific constraints imposed on the system include the uptake of glucose, glutamine, oxygen, glycogen, ketone bodies, leucine, tyrosine, tryptophan, methionine, and ammonia. Since similar proliferative propensities were observed for both glioblastoma (GBM) and medulloblastoma (MB) cell lines, in 5 mL of glutamine for 10 days [92], 0.080 mmol/gDW/h used for iMS570g was employed as the glutamine uptake. Glucose uptake was restricted to 0.852 mmol/gDW/h, which is 11 times higher than glutamine uptake because SHH cells take up 3–11 times more glucose than glutamine [17]. Oxygen uptake was constrained to 0.142 mmol/gDW/h as the optimum value since lower oxygen uptake rates caused a decrease in the PPP flux rates, which is incoherent due to the high activation of PPP in cancer, [14,19] while higher oxygen uptakes increased oxidative phosphorylation activity, which is also contradictory owing to the Warburg effect. The upper bound of leucine uptake was fixed at 0.034 mmol/gDW/h based on the findings that 25 times more glucose is taken up by SHH than leucine [17].

The tryptophan and tyrosine uptake rates of MB over the ones in the healthy cerebellum were determined as 3.5–3.7 and 2.8, respectively [93,94]. Using these ratios and flux rates found in the healthy brain model [15], upper bounds of tryptophan and tyrosine uptakes were computed as 0.0074 mmol/gDW/h and 0.0028 mmol/gDW/h. For methionine uptake, 0.008 mmol/gDW/h in iMS570g model was employed. Lastly, the constraints of ketone metabolism and glycogen uptake were determined as zero because they are only used in case of glucose deficiency and excessive glucose uptake is one of the main features of medulloblastoma [16,91,95]. All constraints are also available in Appendix A.

### 4.2. Operation

All simulations were carried out using CPLEX ILOG optimization algorithms provided by IBM’s Academic Initiative Program on the COBRA toolbox under MATLAB 2017b. The objective function was set as maximization of biomass reaction if not otherwise stated, and the flux balance analysis was conducted after the integration of transcriptome data. Next, the minimization of the metabolic adjustment method (MOMA) was performed to predict closer suboptimal results to the healthy case in order to improve the flux results acquired with FBA [9,96]. The mathematical expression used for MOMA is given.
$$D\;(h,m)\;=\;\surd \;\sum_{i=1}^{n}\;{\left(hi\;-\;mi\right)}^{2}$$

The objective function for MOMA was determined as the minimization of the distance (D) between medulloblastoma (m) and healthy (h) cases. For medulloblastoma (m) and healthy (h) cases, flux rates obtained from FBA in MB and healthy (iMS570) models were utilized. Since this expression includes quadratic terms, quadprog function found in optimization toolbox of MATLAB was employed.

#### Analyses

Flux sampling method was conducted by employing the gpSampler function of the COBRA Toolbox. This function uses Artificially Centered Hit-and-Run (ACHR) algorithm that estimates the random flux profiles [97]. In total, 10,000 flux profiles that satisfy the constraints were attained for each reaction. A two-sample t-test with unequal variances and Benjamini Hochberg Correction procedure were performed on flux sampling results of MB and healthy models to identify flux rates statistically different from each other.

Flux sampling results were also used for the regulation analysis that incorporates flux data with gene expression data by transforming flux and DNA microarray data into statistical scores. Z scores for metabolic fluxes (ZF) and gene expression data values (ZG) were estimated to find the significance of change between MB and healthy conditions.

The reactions whose flux values and/or expression levels of the genes controlling these reactions were higher/lower than 1.96 and −1.96 which were computed with respect to a 5% significance level, were categorized regarding whether they are metabolically or transcriptionally, or post-transcriptionally controlled. ZG scores of some reactions whose controlling genes are more than one were estimated by summing up all ZG scores designated for each gene.

Flux coupling analysis (FCA) was executed with the F2C2 function to uncover reactions that have a direct impact on biomass reactions. Hence, the reactions that can be intervened to hinder growth in MB were revealed. Then, these potential therapeutic targets, which have no destructive impact on healthy cells, were detected.

The essentiality analyses were conducted where all reactions or genes are erased one at a time from the model. If the removal of components leads the flux rate of growth reaction to decline less than the specified flux rate (40%), they are acknowledged as essential. The genes required for energy generation and lactate production along with biomass production rate for the survival of tumor cells were uncovered. Then, the reactions regulated commonly by these essential genes identified in three analyses (biomass, lactate and energy productions set as individual objective functions) were deleted one at a time from the healthy brain and MB models and the genes controlling eliminated reactions that do not influence the basic functions of the healthy cells and decrease the MB growth rate by more than 40% were estimated.

To predict the compounds targeting metabolic enzymes and their impact on both medulloblastoma and healthy cells, the antimetabolites (chemically similar substances to natural metabolites) were explored by using Tanimoto scores and performing FBA. Antimetabolites might diminish the usage of substrates in cells by engendering competitive inhibition; therefore, all the compounds similar to the metabolites present in the medulloblastoma brain model were determined from DrugBank using Simplified molecular-input line-entry system (SMILES) of these metabolites and the compounds with Tanimoto scores higher than 90% were identified [57]. Then, the enzymes catalyzing the reactions in which these metabolites are substrates were inhibited to suppress MB growth. Drug effects on these enzymes were created by lowering constraints of targeted reactions to 0.1 of their flux results detected without inhibition impact in both models. Next, natural metabolites, whose catalyzing enzyme inhibition results in minimum toxic effects on healthy cells and causes a decrease of more than 40% in MB growth, were determined.

## Data Availability

The data that support the findings of this study are available from the corresponding author upon reasonable request.

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
