# Peer review of "Identification of Therapeutic Targets for Medulloblastoma by Tissue-Specific Genome-Scale Metabolic Model"

_molecules, 2023, doi:10.3390/molecules28020779_

Round 1

Reviewer 1 Report

The authors aimed to update the MB metabolic model published by their team. The maisn goal is to better define the metabolic network accounting for MB survival and aggressiveness in order to find biomarkers and specific targets.

A very interesting study is presented, however some points must be clarified to give a correct biological meaning. The authors seem to be more concerned about proving what is already described lacking some updating instead for disclosing new findings and contribute for the evolution of cancer metabolism field. This weakness accounts for the lack of biological meaning and misinterpretations.

The authors claim the Warburg effect is a core pathway in MB. However it is needed to clarify that no longer glycolysis is considered a core energetic pathway in cancer, what is in fact seen is an increased glycolysis rate to sustain biosynthesis. So the upregulation of glucose dependent pathways is observed, as the authors mention regarding pentose phosphate pathway. Accordingly, glutamine dependence is a feature of most malignant cells since glutamine of the main substitute of glucose to supply TCA cycle and OXPHOS. Currently it is set that cancer cells fulfill glycolysis and OXPHOS in parallel, but OXPHOS is not glucose-dependent and the Warburg effect does not represent a metabolic switch but an adjustment of the metabolic flow in order to sustain biosynthesis to feed cell proliferation. It is very important to value Warburg findings by evolving and clarifying his work without being captured in his descriptions and studies. Some studies have pointed this out (doi: 10.3390/cells8030216; doi: 10.1016/j.celrep.2021.109302).

This can explain the authors sentence “Despite low TCA flux rates in MB models owing to the Warburg effect, acetyl-CoA 90 formation in the TCA cycle which is necessary for lipid synthesis is higher compared to 91 the healthy brain model.”… TCA cycle is supplied by substrates other than glucose-derived compounds. The authors present an increased glutamate production in MB models comparing to healthy brain… glutamate derives from glutamine, and originates alpha-ketoglutarate, which enters directly in the TCA, as shown in figure 1. The authors should explore this.

Considering the removal of glucose and the evaluation of its impact in metabolism. The authors consider that without glucose MB cells dispose of a lower energy production. However, it would be interesting to evaluate cancer cell features, such as proliferation, cell death and metabolic viability. Are these cells in fact producing less energy because glycolysis is the main source of energy or the lack of glucose impairs the proliferative capacity of these cells and they enter in a cell death process? Without glucose the glucose-dependent pathways, such as PPP, are shutted down and several biological processes are decreased, without PPP there is no nucleotides to sustain replication and transcription, as well as the redox control is deeply affected. Furthermore, to clarify the glucose dependence for energy production, experiments in the presence of glucose and the absence of glutamine showed be performed. Why this condition was not addressed in Figure 2?

I did not get how the authors conclude this, considering Figure 2, “This result supports the use of the ketogenic diet that aims to deplete the metabolites 112 needed for glycolysis [10].”

Some more rather extrapolated and abusive conclusions are made… “However, while metastasis increased, the ratio of lactate excretion 142 to oxygen uptake rate (LacR/OCR) enhanced slightly, indicating that MB employs its 143 energy for metastasis in preference for growth.” How can the authors conclude this?

“In agreement with FBA and 158 experimental results, 80% of OXPHOS pathway reactions were determined to be 159 significantly different for the neuron in MB relative to the healthy one.” Why? Is the metabolism of glutamine similar in MB and neurons? How can brain specificities considering glutamine explain this? It is very important do find biological explanations.

The authors present evidences without any biological explanation. For instance, “The combination of ACL and FASN genes was captured by the MB model in double gene deletion analysis. In agreement with our simulations, simultaneous inhibition of ACL and FAS genes hindered cancer cell division [45]. Moreover, silencing ACL which is responsible for the production of acetyl CoA in the TCA cycle reduced the growth of lung cancer cells [45]. Suppressing FASN also caused a decrease in hallmarks of tumors like lipid production in various cancers like pancreatic, ovarian, and lung tumors [46].”. Shouldn’t it be expected? ACL produces acetyl CoA to sustain energy and biomass pathways, its abrogation would be damaging. FAs is the unique enzyme synthesizing fatty acids needed for membranes synthesis and processed functioning, off course its abrogation will block cell functioning and survival. How can a cell divide without building blocks?

“The activity of GABA transaminase (ABAT) which controls SuccinateSAL 277 and glutamate productions from alpha-ketoglutarate and GABA is relatively lower in 278 primary MB than in healthy cerebellum [12,47].” Why is that? Which biological specificities underlay this observation? Which normal features are maintained and/or changed in MB cells? How can MB cells benefit from this remodeling?

The fact that lactate can be used as a metabolic source is also ignored in this study. Cancer cells produce high levels of lactate due to glycolysis overactivation, lactate cannot accumulation within the cell and it is exported to be afterwards imported in a controlled manner and used as a carbon and energy source. As it is described in other brain tumors (doi: 10.15252/emmm.202115343).

All the figure legends must be completed to facilitate the results interpretation.

The legend of figure 1 is incomplete…

Figure 1. Main reactions in glycolysis, pentose phosphate pathway (PPP), TCA cycle, and other 74 pathways.

Maybe:

Figure 1. Main reactions in MB are glycolysis, pentose phosphate pathway (PPP), TCA cycle, and other 74 pathways.

Figure 4 is very difficult to follow… the legend must be completed and the colours’ meaning must be presented.

Figure 5… cannot follow the results, cholesterol synthesis is not important as a biomass pathway? What about the cholesterol import?

Figure 7. The relationship between GABA shunt, TCA cycle and electron transfer chain.

This legend is highly incomplete; we should understand the figures by simply reading the legends…

It’s weird to start sentences with numbers “205 reactions out of 594 were estimated to be 151 significantly different from each other according to the p values obtained using a 2-sample 152 t-test with unequal variances. 73% of reactions in the glycolysis pathway in MB showed 153 statistically important deviation from the healthy case because of the increased glucose 154 uptake and excessive lactate production.”

Author Response

Dear Editor Dr. Tijana Milosevic,

Thank you for the review of our manuscript with ID molecules-2091829. Please find below our answers to Referee's Comments, which have helped us clarify and improve our manuscript before resubmitting.

Regards,

K Ulgen (ORCID: 0000-0003-3668-3467)

Reviewer 2 Report

In this work Ilkay Irem Ozbek and Kutlu O. Ulgen present their analysis of Medulloblastoma (MB) using genome-scale metabolic models. They use MB-specific models to identify putative targets and drugs that can be used against MB. The research is not without its merit and some of the findings are quire relevant. However, parts of the manuscript lack clarity and at times it is hard understand what analyses were made and why. Also, language is particularly vague and often inaccurate when referring to metabolic reactions with the authors often using reaction IDs such as R593 and R594 that have little meaning outside of their specific model. I also have some concerns about some of the methods used. These concerns should be addressed before the manuscript is accepted for publications.

Major concerns:

Line 60: It is not clear at all how and why MOMA was used. Normally, one would assume that MOMA is used to simulate the effect of gene KO or equivalent but that does not seem to be the case. Also, the objective used for FBA should be clearly stated as I assume a different objective would be for MB models than the healthy model.

  Line 67: It is unclear why are there 9 MB models, and three instances where transcriptomics data was integrated from GSE

Figure 1: Figure 1 is not informative and does not contribute the paper. I would advise making two separate panels, one for healthy controls and one for with a line width somewhat proportionally to reaction fluxes in each condition. Also, reactions labels are not informative nor defined in the figure foot note. Also, the non-oxidative branch of the pentose phosphate pathway and reductive carboxylation should also be included in the figure or else justify in the footnote why they were not included

Table 1: It is not clear what MB-All, SHH, WNT, GR3 and GR4 are. Neither why all those conditions have exactly the same flux values. Also, it is not clear what is meant by TCA flux given that the TCA cycle has several reactions. Do they mean Pyruvate dehydrogenase?  Related to this, the authors say “ Despite low TCA flux rates in MB models owing to the Warburg effect, acetyl-CoA formation in the TCA cycle, which is necessary for lipid synthesis is higher compared to the healthy brain model” the authors should clarify the source of this acetyl CoA, is it being produced from glutamate?, and if so through which pathway? In this regard, the authors mention “The reaction where glutamate is produced from glutamine showed higher activity in all MB models than the healthy model consistent with the literature where the low expression levels of glutamate uptake proteins, EAAT1 and EAAT2-4 were observed” What is the fate of this glutamate? Is it secreted into the medium or metabolized through the TCA cycle? Also, “The reaction where glutamate is produced from glutamine” should probably be referred as the glutaminase reaction”. Finally, FBA, is notorious for having multiple solutions, the authors should either explore their variation  with FVA or sampling, or use techniques less prone to multiple solutions such as pFBA or else the comparisons in table 1 are meaningless. This is particularly important if the want to use the output of FBA as reference flux for MOMA.

Line 96: “The ratio of glutamate over glutamine concentration was found very close to the data determined experimentally” how was this ratio determined with the GSMMs models? I don’t think it would be correct to use the ratio of the fluxes through glutaminase and glutamine synthase as a surrogate for the concentration ratio.

The paragraph starting in Line 98:

·        “Energy Production without Carbon Source” I hope the authors mean without glucose and/or glutamine; if the models were able to grow or produce ATP without ANY carbon source, it would indicate a severe flaw in their models.

·        The effects of glucose and glutamine deprivation should also be simulated for healthy cells. Showing the effect of deprivation on biomass production would also be useful on assessing whether it would be a good therapeutic strategy against MB.

·        The authors should clarify what they mean by total energy production. Do they mean total ATP production?

·        The authors should explore what is being used as a substrate to produce metabolic energy without glucose and glutamine. Are ketone bodies being used as a substrate for OXPHOS?

Line 112. The authors should expand their reasoning and/or present clear data from their simulations to support their claim. Their discussion should probably also mention that ketone bodies are permeable to the blood brain barrier and physiological substrates of brain under fasting conditions.

Line 144: However, while metastasis increased, the ratio of lactate excretion 142 to oxygen uptake rate (LacR/OCR) enhanced slightly, indicating that MB employs its energy for metastasis in preference for growth. The difference is so small that does appear to be biologically relevant, the authors should abstain from using it to draw conclusions. Even if the differences were significant, I fail to follow the logic of the authors.

Line 164: The reactions whose fluxes deviate significantly from the healthy case but do not have important change in the expression level of their controlling genes are metabolically regulated reactions This is incorrect, such reactions could also be considered as regulated at the posttranscriptional level.

Line 174: In glycolysis, most of the reactions are regulated metabolically (Figure 4). That stems from the fact that a high amount of glucose taken by glucose carriers activates the enzymes regulating the glycolysis pathway. It was detected that lactate synthesis reaction in the glycolysis pathway has an increased ZF score although its ZG score is remarkably low. A high level of pyruvate might be causing an increase in lactate formation.

·        It is hard to believe that the glycolytic enzymes other than glucose transporters are not upregulated when the authors report a tenfold increase on the glycolytic flux (table 1)

·        Enzymes do not regulate the glycolytic pathways, rather they catalyse the reaction of the pathway.  Glucose is the substrate of the pathway so increased glucose uptake can lead to increased flux throught the pathway

·        The lactate synthesis reaction should probably be referred as lactate dehydrogenase

Line 191: Drug Targets for Medulloblastoma. The authors should check that their targets are specific for MB by also testing them on the healthy model. Most of the target they propose seem like they would indistinctly target any cell type so they should prove that they have at least more effect on MB than healthy tissue. This is done in section 3.4.3 but should also be done in section 3.4.2 and 3.4.1 Also, in double gene deletion, lethal gene combinations should probably be referred as synthetic lethal.

Line 345: Antimetabolites Competing with Natural Metabolites; This section is not very clear. Intuitively, the authors should have either simulated the systematic inhibition of every reaction linked to known antimetabolites or simulated the effect of removing every metabolite from the model with the goal of finding new candidates for antimetabolites but its not clear from their description what they did.

Line 216: Figure 4 has the same footnote as Figure 3

I have concerns about the constrains defined in Supplementary Material 1  and SupplementaryMaterial3.m.

·        Why is ketone body uptake set to 0?

·        Why is a uptake of -1000 ammonia allowed, surely the amount of ammonia in brain will be limited enough that amino acids should be the main source a amine group

·         What is the difference between Glucose_A and Glucose_N? (and same for  O2 and glutamine).    

·        Are these constraints used for all simulations or only for model reconstruction? Are they relaxed for simulating gene KO or antimetabolites?

·        Are different constraints used for the healthy model? If so, are most of the changes reported in the paper attributed to different constraints rather than transcriptomics data?

·        Are different constraints used for the mestatic models? If the same constraints are used for all MB models what is the explanation of the following behaviour?: “Due to the higher expression levels of HK2, PFKL, and PKM genes in metastatic amples, the ratio of ATP produced in glycolysis over ATP produced in OXPHOS (ATPG/ATPOP) was detected to be almost 7 times higher in MB-M2 than TPG/ATPOP found in MB-M0”

The genome-scale models used in the simulations should be made available either in the supplementary material or in a public repository

Minor issues:

·        Line 11: Unclear if the authors mean that  “Medulloblastoma (MB) is the most common childhood brain cancer” or that “Medulloblastoma (MB) is the most common childhood cancer in the cerebellum”

·        Line 47: It should probably say something like “over-expression of enzymes involved in lipid  production”

·        Line 49: Warburg effect is not necessarily associated with reduced TCA cycle activity, given that the cells can use alternative substrates such as glutamine

·        Line 53: The authors should clarify if “High Pentose Phosphate Pathway” refers to the oxidative branch, non-oxidative branch, or both of the PPP. Also, in addition to nucleotide production, the oxidative pentose pathway also has major role in redox metabolism by producing reductive power which is  required for biosynthesis and to protect against oxidative stress.

·        Line 56: The authors should expand on the differences between iMS570 iMS570g and the model used in this study

·      ·        Line 84: See previous comments on pentose phosphate pathway

·        Line 149: There multiple flux sampling methods, the authors should state which one they used

·        Line 152: P values should be adjusted for multiple testing

·        Line 154 and Line 158:  The authors should state the total number of reactions with different fluxes in addition to their percentage

·        Line 161: The authors should define what are ZF and ZG.

·        Line 184: Oleoyl-CoA can also be produced from oleic acid, in facto oleoyl-Coa is the metabolically active form of oleic acid

·        Line 193:  such amino acids could be more appropriately referred to as essential AA

Author Response

(The authors gave the same response as above.)

Round 2

Reviewer 1 Report

I have uma more clarification to ask. Concerning point 10, as follow. The authors consider that each cancer cell only expresses MCT4 or MCT1? Can cancer cells, located in an oxygenated area of the tumor and presenting high glycolysis rate to sustain proliferation, manage lactate as a glycolysis end-product to be used aftwards as a metabolic source?

Point 10: The fact that lactate can be used as a metabolic source is also ignored in this study. Cancer cells produce high levels of lactate due to glycolysis overactivation, lactate cannot accumulation within the cell and it is exported to be afterwards imported in a controlled manner and used as a carbon and energy source. As it is described in other brain tumors (doi: 10.15252/emmm.202115343).

Response 10: Solid tumors are characterized by their metabolic heterogeneity. The metabolic needs of tumor cells change based on their location (Whether they are located in the edge or center as indicated in these studies (doi: 10.3390/ijms21176254 and doi: 10.3389/fonc.2019.01143). As the reviewer suggested, there is a symbiotic relationship between glycolytic and oxidative cells found in the tumor environment. While glycolytic cells upregulate MCT4 that exports lactate, oxidative cells express MCT1 transporter which imports lactate. To reflect the heterogeneity of the tumor, it is necessary to create at least two separate tumor cells in different locations. Additionally, for tumor cells located in different locations (Edge or center), different constraints and different transcriptome data should be used to mimic their metabolic activities in the computational model. However, finding the constraints (the uptakes of substances internalized by MB cells) measured experimentally and specifically for medulloblastoma was a very challenging process because of the lack of experimental data. While it would be very useful to design this kind of model that reflects the symbiotic relationship for lactate in different locations of the tumor, it would also be out of the focus of this study. Still, our model includes lactate exchange reactions which export most of the lactate to extracellular space to sustain continuously high rates of glycolysis and decrease intracellular acidification. 

Author Response

Dear Editor Dr. Tijana Milosevic,

Thank you for the second review of our manuscript with ID molecules-2091829. Please find below our answers to Referee's Comments-Round 2, which have helped us clarify and improve our manuscript before resubmitting.

Regards,

K Ulgen (ORCID: 0000-0003-3668-3467)

Reviewer 2 Report

Overall I find that all of my major concerns with the manuscript have been addressed. I believe that the manuscript should be accepted, provided the following minor concerns are addressed:

The paragraph starting in line 263: I refer the authors to the work by Zur et al (https://doi-org.ezp.lib.cam.ac.uk/10.1093/bioinformatics/btq602 and similar works by the same authors such as https://doi-org.ezp.lib.cam.ac.uk/10.1038/s41598-019-54221-y ) and particularly to the following paragraph

The correlation between expression and metabolic flux is generally moderate and in some cases significant transcriptional changes do not reflect changes in flux, and vice-versa, significant changes in measured flux may not reflect transcriptional changes (Ovacik and Androulakis, 2008). These discrepancies may result from post-transcriptional regulatory processes that effect the actual levels of enzymes translated and from metabolic regulation, representing the effect of metabolite concentrations on the actual enzyme activity through allosteric and mass action effects (Rossell et al., 2006).

Simply put, as the authors have only gene (transcript) expression data they can identify transcriptionally regulated reactions as reactions where transcripts and fluxes have consistent changes , and reactions not regulated at the transcriptional level )(discrepancy between fluxes and transcripts). However, I believe they lack data to distinguish between metabolically regulated reactions and post-transcriptionally regulated reactions. For instance, fluxes deviate significantly from the healthy case but do not have an important change in expression level of their controlling genes might indeed be driven by metabolic regulation but they might also be driven by post-transcriptional regulation of total enzyme concentration.

Line 142: They should include in the main manuscript some the additional context provided in the answer to reviewers files (i.e state that the ratio of glutamate over glutamine accumulation predicted by their model was found to be very close to the concentration ratio determined experimentally)

Author Response

(The authors gave the same response as above.)
